# Review of psychological effects of dog bites in children

Carri Westgarth [iD],[1] Serena Provazza,[2] Jade Nicholas,[3] Victoria Gray[2]

[1]Department of Livestock and One Health, Faculty of Health and Life Sciences, University of Liverpool, Neston, UK
[2]Department of Clinical Health Psychology, Alder Hey Children's NHS Foundation Trust, Liverpool, UK
[3]Independent Researcher, Hampshire, UK

**Correspondence to**
Professor Carri Westgarth; Carri.Westgarth@liverpool.ac.uk

## ABSTRACT

**Background** Dog bites are a concerning health problem in children and one of the leading causes of non-fatal injuries in this population. Dog attacks not only cause physical injuries but can also lead to long-term psychological problems. A review was performed to investigate the scope of literature on the psychological effects of dog bites on a paediatric population.

**Methods** A literature search was performed on Web of Knowledge database between 1982 to June 2023, returning 249 results. 14 primary studies reporting the psychological consequences of dog bites in children or adolescents were classed as eligible and 9 further studies were added from prior knowledge and bibliographical searches. 23 studies involving 1894 participants met the criteria and were included in this review.

**Results** Of these 23 studies, 8 were case studies or small case series reports (up to n=4), 14 larger descriptive studies and 1 analytical cross-sectional study. There was a mixture of retrospective and prospective data-gathering. The most common psychological consequences of dog bites in children were post-traumatic stress disorder, dog phobia, nightmares and symptoms of anxiety and avoidance behaviours.

**Conclusions** Studies on dog bites in a paediatric population with a specific focus on the psychological consequences associated with dog bites and their management are sparse. Future research and practice should more greatly consider the psychological impact on child victims of dog bites and their family members, as well as their management to avoid the development of mental health issues and improve their quality of life. Future research also needs to ascertain the efficacy of using virtual reality in treating children with dog phobia.

---

## KEY MESSAGES

⇒ Psychological consequences of dog bites on children are common but under-researched and often overlooked.
⇒ Typical consequences of dog bites are post-traumatic stress disorder, dog phobia, nightmares and symptoms of anxiety and avoidance behaviours.
⇒ Treatment of dog phobias can be challenging to deliver.
⇒ Virtual reality offers a potential ethical and safe, controlled environment for cognitive behavioural therapy regarding dogs.

---

It is well established that most of the child dog bite victims are bitten inside the home, and most dogs involved are known to the victim.[6 7] The COVID-19 global pandemic and the subsequent implementation of lockdowns with a 'stay at home' order forced children to spend more time than usual at home with a significant increase in dog bites.[8 9]

Some evidence supports that children are more vulnerable to dog attacks compared with adults because of their smaller physical size, their underestimation of risk and their tendency to behave more impulsively,[10] as well as leaning-in behaviour with animals in young children.[11] Rates of admission peak in the 5–9 age group.[8] Furthermore, children are often injured in the neck and head regions and many have facial wounds.[12–14] The younger the child the more frequent the bites are to the face[14 15] and young children have more extensive/severe injuries.[13 16]

In addition to physical impact, dog bites often carry healthcare costs.[8] They also produce psychological costs to the victims and other parties involved,[17–19] however, compared with physical, psychological consequences of dog bites and their management is still scarcely investigated and poorly reported[20 21] Timely recognition of psychological symptoms caused by dog attacks in children and subsequent psychological support to the child victims and their families would be of critical importance to prevent development of future mental health issues,

## INTRODUCTION

Dog ownership is argued to provide some health benefits,[1–3] however, dogs, as well as other animals, may cause severe injuries to humans. Although there are no global estimates of dog bites incidence, the WHO reports dog bites account for tens of millions of injuries annually.[4] As a result, dog bites represent a major public health concern, affecting predominately the paediatric population.[5] In the UK, the incidence of hospital admissions for dog bites in children (14 or under) is stable at approximately 14.4 per 100 000 population per year.[5]

which would in turn influence the quality of life of the children and their caretakers. The aim of this systematic review was to explore current evidence on the psychological impact of dog bites in children and adolescents.

## METHODS
### Protocol and inclusion criteria
This review was conducted according to the standards established by the Preferred Reporting Items for Systematic Reviews.[22]

A search of Web of Science Core Collection database was performed to identify original research articles focusing on psychological consequences of dog bite injuries in the paediatric population from 1982 to June 2023. This single popular database was chosen due to time constraints and its broad coverage of science, social sciences and arts and humanities. The search term used on all fields was:

('child* OR pediatric') AND (dog* OR cani*) AND (bit* OR attack*) AND (psychological OR PTSD OR phobia OR trauma* OR anxiety)

At least the abstract needed to be available in English for assessment for relevant inclusion, and screening was performed by CW and SP. Articles were excluded if: (1) they were not an original primary source of evidence in a scientific journal (ie, book chapter or review papers were excluded); (2) the study population did not include children (ie, under 18 years of age, although it could also include adults); (3) studies focused on bites in general (eg, human bites, cat bites) rather than dog; (4) the information reported was not specific to psychological effects of dog bites but rather on dog bites in general (eg, wound management, dog bite incidence). Articles were excluded based on reading the full paper if the title or abstract did not already clearly exclude it; if the study purpose and population were deemed relevant, often the full paper needed to be read in order to find if psychological effects had been reported. Further relevant studies were also identified from bibliographic searches and prior knowledge. See figure 1 for the study search and inclusion flowchart. Relatively few studies met the inclusion criteria and thus sample size was low. In addition, the study designs found would be classed as low-level of evidence (such as case studies, descriptive case series and cross-sectional studies). For these reasons, all studies and their findings have been reported here rather than excluded based on quality. Evaluation and summary of the study design and psychological findings was performed by CW and JN.

## RESULTS
Out of 249 papers screened for eligibility, 14 studies met the criteria and were included in this review, along with 9 other studies identified through prior knowledge and bioblographies, totalling 23 studies involving 1894 participants (figure 1). Of these 23 studies, 8 were single case studies or small case series (up to four children per report)[23–30] (online supplemental table 1). 13 were descriptive studies of larger patient groups,[7 15 18 19 31–39] 1 was a case-control study but the psychological findings were only descriptional[40] and 1 was an analytical cross-sectional study[41] (online supplemental table 2 and 3). A mixture of retrospective (ie, incidental review of past case records, a survey regarding fear of dogs, see online supplemental table 2) and prospective data collection (deliberate interview of dog bite cases as they attended hospital, follow-up interview of a cohort of bitten children, see online supplemental table 3 methods were reported depending on the study, and sometimes both.

### Effects on children
One study specific to dog bite patients estimated that 70% of children had demonstrated concerning behaviours since the incident.[37] Post-traumatic stress disorder (PTSD) was mentioned in eight studies: two of four children (a family in which one child was bitten),[23] diagnosed in a case study,[28] diagnosed in descriptive studies of dog bite patients at a prevalence of 12/22,[31] 12/22,[19] 19/358[18]; 98% within 5 days of a traumatic facial incident some of which were dog bites,[35] and also mentioned more generally in two other studies, one which included (16%) child dog bite patients[34] and one specifically of child dog bites.[36] PTSD was typically diagnosed through the use of a screening questionnaire or meeting six major criteria under DSM-IV (Diagnostic and Statistical Manual of Mental Disorders, fourth edition). Acute stress disorder (ASD) was also mentioned (38/358 patients in a study by Ji *et al*[18]) and observed to be an early indicator of PTSD. In a study of children interviewed after a traumatic experience (16% of which were dog bites), PTSD symptoms were greater in the dog attack and parental violence groups than the 'mild stressor' group, and mothers of bitten children were also observed to develop PTSD which was associated with child PTSD score as reported by the parent.[34] In addition, in the dog attack group mothers observed girls to have greater PTSD symptoms and boys to have greater dissociative symptoms, while child self-report showed boys to have greater PTSD symptoms. Other studies did not find evidence of gender/age effects in regards to PTSD in child dog bites[18 31] The severity of the symptoms seemed to be related to the severity of the attacks; children who sustained more physically severe or multiple bites were more likely to develop PTSD symptoms.[18 19 31]

Other than formally reporting PTSD/ASD, specific symptoms most commonly described in the articles were fearfulness/avoidance of dogs[24 25 27 30 33 36–39] and sleep disturbance/nightmares.[23–25 31 36–38 40] Other symptoms included bed wetting,[23 25] selective mutism,[28] traumatic memories/re-living,[19 28 31] talking a lot about the incident,[37] increased arousal,[19 31] anger/aggression,[24 31] withdrawn/depressed/numbing,[19 28 31 35] fearful,[37] panic attacks,[27] anxiety,[35 36] hypervigilance,[31] difficulty playing/avoidance of playing outdoors,[24 30] anxiety about

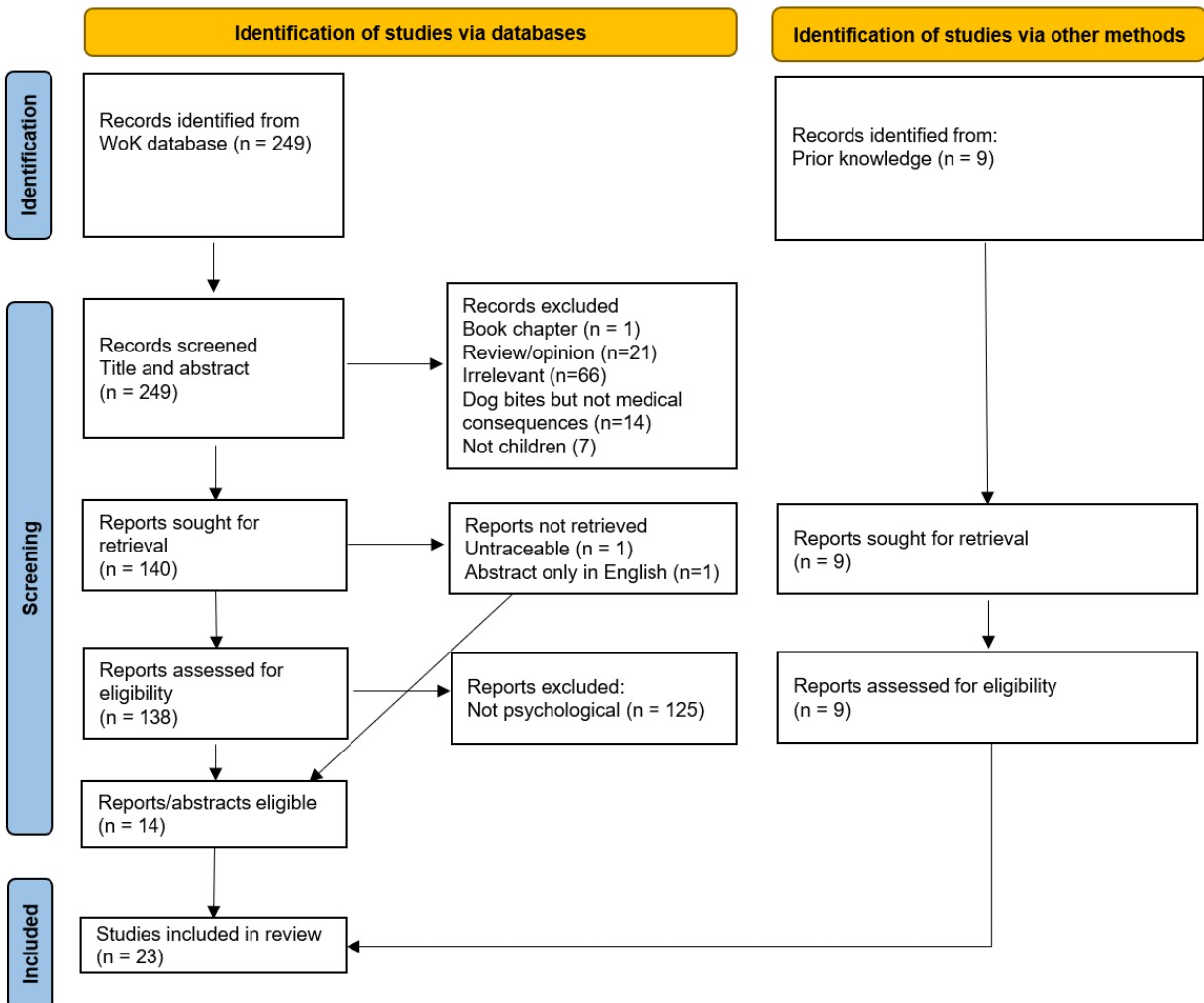

**Figure 1** Flow chart of study search and screening process for studies of psychological consequences of dog bites on children.

hospitals,[37] fear of dying or re-injury[35] and phobia of own image.[36]

### Effects on parents

Notably, parents reported changes in their behaviours as well, proving that dog bites represent a burden not only for the victims but also for their families.[34 37] Parents described feeling shaken,[25] guilty, fearful for their child's safety and worried about scars.[37]

### Treatment of children

Despite the reporting of psychological symptoms after dog bites in children, it was rare for psychological treatment to be recorded; although it featured in case studies,[23 25 27–30] when evaluating descriptive studies it was only reported in 2/100,[15] 2/277,[7] 2/38[40] patients. Indeed, two studies specifically noted no children received any psychological services (0/22[19]; 0/34[37]) but 50% of parents felt it would be helpful.[37]

### Other observations

Hon *et al* acknowledged that one challenge to collecting these data is that in an emergency department service where dog bites present is it might be difficult to gather information about psychological and emotional trauma.[21] Clearly, there is an unidentified gap in both the literature and patient need for psychological assessment. In addition, it has been suggested that interventions should also include educational programmes on the risk and severity of dog attacks offered to parents and children, given dog bites can have significant psychological consequences that can negatively impact children's quality of life.[29]

### DISCUSSION

The main purpose of this review was to explore the current literature on paediatric dog bites, with a

specific focus on the psychological impacts of dog bites in children, as the psychological consequences of dog bites as well as their treatment are often overlooked.[20] [21] Evidence suggests that children bitten by dogs are at high risk of developing psychological disturbances, ranging from fear of dogs to clinical PTSD, however, few studies mentioned psychological management or treatment offered after dog attacks. It appears that assessing children in an emergency department (ED) to evaluate psychological sequelae following dog bites, and further offering them psychological support, can be difficult and/or lacks consideration. This may result in psychological symptoms significantly impacting children and potentially worsening. A form of follow-up assessment is likely to be required to assess whether initial symptoms of psychological distress have settled, but whom this responsibility falls to once a patient is discharged from the ED, is unclear. Further, not only did dog injuries have an impact on the victims, but they also affected their carers. Psychological symptoms shown by young victims of dog bites seemed to be long-lasting with some studies reporting over 12 months. Age and gender differences were inconclusive between studies but across studies more severe bites commonly resulted in more serious psychological impact.

It has become increasingly recognised that there is a need and opportunity to evaluate victims of traumatic injuries for psychological distress so that they can be appropriately referred on. For example, this can occur during physical treatment such as plastic surgery.[42] However, it is clear from our review that many children who have been bitten by dogs or are otherwise fearful of dogs may benefit from psychological screening and if required, treatment, but this does not appear to be regularly occurring in practice. This is particularly concerning given children are in a crucial developmental stage with varied levels of supportive systems and resilience, and PTSD is a common psychiatric disorder after a child has a traumatic experience.[43]

Although different approaches have been found to be effective in treating results of trauma and in particular phobia of dogs in children, CBT (cognitive behavioural therapy) regarding some form of graded exposure appears to be the most widely used.[44] [45] A difficulty with this is the ethical and practical safety implications regarding controlled exposure to real animals who can be difficult to access, unpredictable and to some extent uncontrollable in their behaviours, and with their own welfare considerations.[46] For example, one intervention states 'Parents were not provided with explicit instructions on how to find dogs but were encouraged to consider various options: dogs owned by extended family members, neighbors, and friends; dogs at pet stores, breeders, and animal shelters; dogs at the park or other recreation areas'.[45] Another play-based intervention used dogs with handlers within the intensive session, with no description about how these had been assessed for suitability, and 'during the session, the child was encouraged to engage in a range of different tasks with each dog including approaching and patting the dog, offering the dog treats and walking the dog on the lead'.[44] Great care must be taken in order to be able to gradually control the intensity of exposure and also safeguard both child and animal welfare during such treatment so that neither becomes overwhelmed and both have a positive experience. This includes providing very clear guidance to those delivering it, including parents.

A potential more ethical solution to these challenges during the intensive and early stages of therapy would be to use virtual reality (VR).[46] This would allow a gradual and controlled increase in exposure to particular dog behaviours required for that case. For example, Farrel *et al* have suggested the use of virtual reality therapy in treating dog phobia can be possible using just one treatment session.[47] In eight children with a specific phobia of dogs, after 1 month follow-up, 75% of the children were considered recovered. The authors concluded that VR can be effectively used as an alternative to the classic in-vivo exposure-based therapy and might overcome some of its challenges as the most difficult CBT technique to deliver. More research is needed to confirm the efficacy of VR as an effective treatment of dog fear in a paediatric population.

This review has the merit of investigating and summarising existing evidence on psychological effects of paediatric dog bites and their management, as to the best of our knowledge, such investigation has not been conducted before. However, it presents with some limitations. First of all, even though the literature search was performed on a highly relevant database, it was only a single database and some studies may be missing from this review. Additionally, of the few studies identified, many had a very limited sample size, which may have limited the statistical power and generalisability of the findings. They also mostly used study designs with a high risk of bias, such as case studies, and descriptive cross-sectional studies, often with retrospective data collection simply by reviewing case notes. For example, unless it was particularly noted because a parent or child raised the issue, psychological impacts may have remained unreported. Those studies of a potentially lower risk of bias that used prospective data collection still varied in the quality of screening methods for psychological outcomes and whether follow-up was used. Therefore, the conclusions drawn here should be taken with caution. Notwithstanding the above limitations, this review provides an insight into the psychological consequences of paediatric dog bites.

To conclude, dog bites in children represent a traumatic event that can cause devasting psychological consequences in the victims and in their families. A thorough investigation of the psychological impact of dog bites on children and their parents followed by a prompt multidisciplinary management of both physical and psychological symptoms would lead to better outcomes, preventing the occurrence of more severe mental health problems such as phobia and PTSD and improving the quality of life of the victims and their families. Furthermore, educational programmes on the risk and severity of dog attacks should be offered to parents and children, to prevent dog bites. CBT can be used to treat the fear of dogs in children, with VR representing an alternative to the classic in vivo exposition therapy that requires more investigation. Future research should focus more on the psychological impact of dog injuries and on the treatment of child victims of dog bites and their family members, to avoid the development of mental health issues and improve their quality of life.

**Contributors** CW conceptualised the paper. CW and SP performed the searches and paper selection. CW and JN reviewed the papers and assimilated the data. CW and SP wrote the first draft. All authors including VG commented on and revised the paper. CW submitted the paper. CW is responsible for the content as guarantor.

**Funding** This work was supported by the Faculty of Health and Life Sciences, University of Liverpool.

**Competing interests** CW is a member of the Merseyside Dog Safety Partnership, which has a website containing useful resources for dog bite prevention. www.merseydogsafe.co.uk. There are no other competing interests to declare.

**Patient consent for publication** Not applicable.

**Ethics approval** Not applicable.

**Provenance and peer review** Commissioned; externally peer reviewed.

**ORCID iD**
Carri Westgarth http://orcid.org/0000-0003-0471-2761

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
