## [Reviewer comments · BMJ Paediatrics Open]

ARTICLE DETAILS

TITLE (PROVISIONAL)	
AUTHORS	Westgarth, Carri Provazza, Serena Nicholas, Jade Gray, Victoria

VERSION 1 – REVIEW

REVIEWER	Dr. Richard Armitage University of Nottingham Division of Epidemiology and Public Health, University of Nottingham, Nottingham NG7 2RD, UK Division of Epidemiology and Public Health Nottingham NG7 2RD United Kingdom of Great Britain and Northern Ireland
REVIEW RETURNED	04-Apr-2024

GENERAL COMMENTS	Many thanks to the authors for their work, and for the opportunity to review it. The authors conducted a systematic review to explore the current evidence on the psychological impact of dog bites in children and adolescents. The paper addressed an important and seemingly under-researched area. I found the work to be well written and am generally in favour of it's being published if the following points can be satisfactorily addressed. 1) Only one database was searched. Please could the authors justify this decision and consider searching additional databases of medical literature.2) The authors should assess and comment on the quality/certainty/confidence of the studies that were included in the review, including their risk of bias, and adjust their discussion and recommendations to reflect this.3) PTSD is a formal psychiatric disorder but is increasingly used in a more informal manner. Did the studies explain how this diagnosis was reached? Please comment on this.4) Do the authors have any insight into the prevalence of dog bites in children that did not lead to negative psychological impacts? Commenting on this would provide useful context and guard against over-estimating the psychological impacts on children.5) In the concluding paragraph, the authors should take care to accurately reflect the efficacy of VR in this context, which is unclear (the authors rightly call for more research in this area earlier in the paper and should do the same in the conclusion).
--

REVIEWER	Chris Mannion Leeds Teaching Hospitals NHS Trust
-----------------	---

	United Kingdom of Great Britain and Northern Ireland
REVIEW RETURNED	21-Apr-2024

GENERAL COMMENTS	Thank you for this. It is an important topic. Trauma and traumatic events are very important in childhood development. Firstly, there are a few spelling errors noted which should be addressed please. For example the opening 5 words of this paper reads "Dog bites are a concerning heath problem..." More generally, I think this is a helpful contribution to the literature and at least shines a light on this under reported area. As this is a systematic review of the psychological effects of dog bites in children, I wonder if in the discussion (with a least albeit a minor mention) there is some recognition given to the effects and impact of traumatic injuries. Some parallels to the manner in which the post traumatic sequela is approached/addressed may be helpful? The notion of VR to address dog phobia and thus help with this PTSD treatment could prove helpful. That said, I am happy to accept. Thank you.
---

VERSION 1 – AUTHOR RESPONSE

We would like to thank all reviewers and editors for their very helpful feedback.

1) Author Names

Please check on the Author's names in the main document and in the system below. The names indicated in the main document must match the name registered in the ScholarOne submission system.

System: Westgarth, Carri
Provazza, Serena
Nicholas, Jade
Gray, Victoria

Main Document: Westgarth C.
Provazza, S.
Nicholas, J.
Gray, V.

Response:

CORRECTED

2) Different Funding Statement

Upon checking your manuscript, I noticed that the Finding Statement in the main documents and funder listed in the system is different. Kindly update your records and ensure that all data provided

in the system should be matched in your main document file.

Response:

CORRECTED

3) Contributorship Statement

Please provide a clear and specific Contributorship statement. This must include all authors' names or initials to have been mentioned at least once. Kindly provide this statement in your main document and ensure that it is identical to the statement provided in the ScholarOne submission system.

The Contributorship statement should make clear which individual author has contributed to a task. This may discuss the planning, conduct, reporting, conception and design, acquisition of data or analysis and interpretation of data, etc. It should also identify one, or occasionally more, contributor(s) as being responsible for the overall content as guarantor(s).

Response:

We had a contributor statement already but I have added 'CW is responsible for the content as guarantor' and added 'including VG' in one part.

4) Table exceeds 3 pages or has more than 9 columns

Your table 1 and 2 exceed two more pages. If the table is more than 2 pages long or has more than 9 columns, this has to be removed from the main document and designated as Supplementary table instead.

Response:

Moved to supplementary material.

5) Missing Grant Number

You have indicated a funder/s for your paper. Please ensure to provide an award/grant number for your funder/s in the submission system. If the funder cannot provide an award/grant number, you can indicate N/A for the award/grant number.

Response:

N/A added.

Editor(s)' Comments to Author (if any):

Agree minor revisions required. Also please follow author guidelines and provide a structured Abstract. BMJPO requires a structured abstract of up to 300 words, with following subtitles:

Background; Methods; Results; Conclusions.

Response:

Corrected

Associate Editor

Comments to the Author:

Thanks for submitting this excellent review. Minor comments from both reviewers.

Title add "a narrative review"

Response:

The abstract said 'systematic review' due to the process we followed, so I am unsure about adding 'narrative' to the title as it is more than that. However I appreciate that this is arguably not a full systematic review because of the more limited search strategy, lack of assessment of risk of bias and no meta-analysis. Therefore I have added 'A review of..' to the title and taken 'systematic' out of the abstract and methods.

Table 2 would be better as two tables divided into prospective and retrospective studies, ie one table for each type of study. (Possibly divided into three tables: prospective;retrospective;both)

Response:

Have split into two and moved tables to supplementary material as requested by the editor as still too long.

Results Summary of findings. Add subheading for "Effect on parents".

Response:

For consistency I have added subheadings 'effect on children', effects on parents' and 'other observations'.

Consider adding a table listing the most common psychological problems identified

Response:

Thanks for this suggestion which is an interesting one. It is difficult defining 'common' when all we have to go on is whether a study mentioned it rather than a consistent specific screening of patients for all psychological outcomes.

We have the below already covered in the text as our interpretation of what we have read and in addition don't want to be too repetitive?

"Other than formally reporting PTSD/ASD, specific symptoms most commonly described in the articles were fearfulness/avoidance of dogs 24 25 27 30 33 36-39 and sleep disturbance/nightmares 23-25 31 36-38 40. Other symptoms included bed wetting 23 25, selective mutism 28, traumatic memories/re-living 19 28 31, talking a lot about the incident 37, increased arousal 19 31, anger/aggression 24 31, withdrawn/depressed/numbing 19 28 31 35, fearful 37, panic attacks 27, anxiety 35 36, hypervigilance 31, difficulty playing/avoidance of playing outdoors 24 30, anxiety about hospitals 37, fear of dying or re-injury 35, and phobia of own image 36."

However if you prefer us to move this into a table along with the PTSD/ASD references we can do, but I am aware that might not leave a lot in the 'results' section of the text?

Reviewer: 1

Dr. Richard Armitage, University of Nottingham

Comments to the Author

Many thanks to the authors for their work, and for the opportunity to review it.

The authors conducted a systematic review to explore the current evidence on the psychological impact of dog bites in children and adolescents.

The paper addressed an important and seemingly under-researched area.

I found the work to be well written and am generally in favour of it's being published if the following points can be satisfactorily addressed.

Response:

Thank you for your kind words and we are glad you enjoyed the paper.

1) Only one database was searched. Please could the authors justify this decision and consider searching additional databases of medical literature.

Response:

This was an invited narrative review and conducted mostly in our 'spare' time. Nevertheless we wished to be as systematic as we could in our approach to it. Broad search terms needed to be used due to the topic of interest, as it could be that psychological findings were reported in studies of

child dog bites even though not mentioned in the title, keywords or abstract. Therefore even with one database there would be a considerable number of papers to search in full, extending the search time. Therefore for practical reasons we chose to only search one key database.

We have modified and justified to” A search of Web of Science Core Collection database was performed to identify original research articles focusing on psychological consequences of dog bite injuries in the paediatric population from 1982- June 2023. This single popular database was chosen due to time constraints and its broad coverage of science, social sciences and arts and humanities.”

We have also expanded the sentence in the limitations section:

“First of all, even though the literature search was performed on a highly relevant database, it was only a single database and some studies may be missing from this review.”

2) The authors should assess and comment on the quality/certainty/confidence of the studies that were included in the review, including their risk of bias, and adjust their discussion and recommendations to reflect this.

Response:

In the methods we had previously stated that “Due to the low number of studies that met the inclusion criteria and low-level of evidence study designs used (case studies and descriptive case series), all studies have been reported here rather than exclusion based on quality.”

We have modified this for clarity to:

“Relatively few studies met the inclusion criteria and thus sample size was low. In addition, the study designs found would be classed as low-level of evidence (such as case studies, descriptive case series and cross-sectional studies). For these reasons all studies and their findings have been reported here rather than exclusion based on quality.”

We have also added more discussion of this in the discussion section:

“They also mostly used study designs with a high risk of bias, such as case studies, and descriptive cross-sectional studies, often with retrospective data collection simply reviewing case notes. For example, unless it was particularly noted because a parent or child raised the issue, psychological impacts may have remained unreported. Those studies of a potentially lower risk of bias that used

prospective data collection still varied in quality of screening methods for psychological outcomes and whether follow-up was used. Therefore, the conclusions drawn here should be taken with caution.”

Finally, the now splitting of the retrospective and prospective data studies into table 2 and 3 may assist the reader with this interpretation.

3) PTSD is a formal psychiatric disorder but is increasingly used in a more informal manner. Did the studies explain how this diagnosis was reached? Please comment on this.

Response:

We have added to results:

“PTSD was typically diagnosed through use of a screening questionnaire or meeting six major criteria under DSM-IV”

4) Do the authors have any insight into the prevalence of dog bites in children that did not lead to negative psychological impacts? Commenting on this would provide useful context and guard against over-estimating the psychological impacts on children.

Response:

As mentioned previously this is difficult as most of the studies were not well set up to measure prevalence as they didn't specifically screen for psychological outcomes. When they did, they varied as to what they were screening for, for example may screen for full PTSD but not fear of dogs, and 'psychological effects' are a broad term. Further, it depends when you ask as to what the prevalence is, for example one study said 98% at 5 days post-incident were symptomatic for PTSD and 44% at one year with 21% meeting diagnostic criteria. The studies also vary in what intensity of trauma the patients had experienced, for example general dog bite to intensive facial surgery. For this reason we have refrained from trying to summarise overall prevalences across studies in the text as feel uncomfortable with this, and instead reported individual findings in the tables. Even so, we aren't talking about the minority here with the types of percentages being reported within the individual studies when more intensive screening does occur.

5) In the concluding paragraph, the authors should take care to accurately reflect the efficacy of VR in this context, which is unclear (the authors rightly call for more research in this area earlier in the paper and should do the same in the conclusion).

Response:

We have modified this sentence to

“Cognitive behavioural therapy can be used to treat fear of dogs in children, with VR representing an alternative to the classic in vivo exposition therapy that requires more investigation”

Reviewer: 2

Chris Mannion, Leeds Teaching Hospitals NHS Trust

Comments to the Author

Thank you for this. It is an important topic. Trauma and traumatic events are very important in childhood development.

Firstly, there are a few spelling errors noted which should be addressed please. For example the opening 5 words of this paper reads "Dog bites are a concerning health problem..."

Response:

Thanks, we have checked and corrected a few errors.

More generally, I think this is a helpful contribution to the literature and at least shines a light on this under reported area. As this is a systematic review of the psychological effects of dog bites in children, I wonder if in the discussion (with at least a minor mention) there is some recognition given to the effects and impact of traumatic injuries. Some parallels to the manner in which the post traumatic sequelae is approached/addressed may be helpful?

Response:

Thanks. We have expanded in the discussion to:

“It has become increasingly recognised that there is a need and opportunity to evaluate victims of traumatic injuries for psychological distress so that they can be appropriately referred on. For example this can occur during physical treatment such as plastic surgery.⁴² However, It is clear from our review that many children who have been bitten by dogs or are otherwise fearful of dogs may benefit from psychological screening and if required, treatment, but this does not appear to be regularly occurring in practice. This is particularly concerning given children are in a crucial

developmental stage with varied levels of supportive systems and resilience, and PTSD is a common psychiatric disorder after a child has a traumatic experience.⁴³

Although different approaches have been found to be effective in treating results of trauma and in particular phobia for dogs in children, CBT (cognitive behavioural therapy) regarding some form of graded exposure appears to be the most widely utilised ^{44 45....}”

The notion of VR to address dog phobia and thus help with this PTSD treatment could prove helpful. That said, I am happy to accept. Thank you.

Response:

Thank you for your kind review.